# Augmentative Pharmacological Strategies in Treatment-Resistant Major Depression: A Comprehensive Review

**DOI:** 10.3390/ijms222313070

**Published:** 2021-12-02

**Authors:** Alice Caldiroli, Enrico Capuzzi, Ilaria Tagliabue, Martina Capellazzi, Matteo Marcatili, Francesco Mucci, Fabrizia Colmegna, Massimo Clerici, Massimiliano Buoli, Antonios Dakanalis

**Affiliations:** 1Psychiatric Department, Azienda Socio Sanitaria Territoriale Monza, 20900 Monza, Italy; e.capuzzi1@campus.unimib.it (E.C.); m.marcatili@asst-monza.it (M.M.); f.colmegna@asst-monza.it (F.C.); massimo.clerici@unimib.it (M.C.); 2Department of Medicine and Surgery, University of Milano Bicocca, 20900 Monza, Italy; i.tagliabue5@campus.unimib.it (I.T.); m.capellazzi@campus.unimib.it (M.C.); antonios.dakanalis@unimib.it (A.D.); 3Department of Medicine and Surgery, University of Milan, 20122 Milan, Italy; francescomucci@hotmail.it; 4Department of Pathophysiology and Transplantation, University of Milan, 20122 Milan, Italy; massimiliano.buoli@unimi.it; 5Department of Neurosciences and Mental Health, Fondazione IRCCS Ca’ Granda Ospedale Maggiore Policlinico, 20122 Milan, Italy

**Keywords:** augmentation, treatment-resistant depression, psychopharmacology, review

## Abstract

Treatment resistant depression (TRD) is associated with poor outcomes, but a consensus is lacking in the literature regarding which compound represents the best pharmacological augmentation strategy to antidepressants (AD). In the present review, we identify the available literature regarding the pharmacological augmentation to AD in TRD. Research in the main psychiatric databases was performed (PubMed, ISI Web of Knowledge, PsychInfo). Only original articles in English with the main topic being pharmacological augmentation in TRD and presenting a precise definition of TRD were included. Aripiprazole and lithium were the most investigated molecules, and aripiprazole presented the strongest evidence of efficacy. Moreover, olanzapine, quetiapine, cariprazine, risperidone, and ziprasidone showed positive results but to a lesser extent. Brexpiprazole and intranasal esketamine need further study in real-world practice. Intravenous ketamine presented an evincible AD effect in the short-term. The efficacy of adjunctive ADs, antiepileptic drugs, psychostimulants, pramipexole, ropinirole, acetyl-salicylic acid, metyrapone, reserpine, testosterone, T3/T4, naltrexone, SAMe, and zinc cannot be precisely estimated in light of the limited available data. Studies on lamotrigine and pindolol reported negative results. According to our results, aripiprazole and lithium may be considered by clinicians as potential effective augmentative strategies in TRD, although the data regarding lithium are somewhat controversial. Reliable conclusions about the other molecules cannot be drawn. Further controlled comparative studies, standardized in terms of design, doses, and duration of the augmentative treatments, are needed to formulate definitive conclusions.

## 1. Introduction

Major depressive disorder (MDD) is one of the most common psychiatric disorders and is associated with work/social dysfunction, accounting for a significant financial burden worldwide [1]. Among individuals receiving an adequate pharmacological treatment, only 30% reach a full symptom recovery. The remaining 70% of MDD patients will experience either a pharmacological response without remission or no response at all [2], thus configuring treatment resistant depression (TRD).

TRD is a complex phenomenon, influenced by different factors, including advanced age, comorbid psychiatric disorders, and medical illness [3]. There is no consensus on the definition of TRD. According to the European Medicine Agency, TRD patients are those who fail to show clinically meaningful improvement after treatment with at least two antidepressant (AD) agents of different classes, prescribed in adequate dosages for an appropriate period of time and with adequate affirmation of treatment adherence [4]. This definition is in agreement with the World Federation of Societies of Biological Psychiatry (WFSBP) Guidelines [5] about unipolar depressive disorder. Of note, Thase and Rush [6] proposed a classification of TRD staging, which was cited in several studies and is currently accepted in the literature on this topic. However, a recent systematic review by Brown and collaborators [7] highlighted that only 48.4% of the published literature specified TRD as the failure of at least two treatments. Despite the assessment of TRD having improved over the last few years, the lack of consensus may limit the ability of researchers to generalize the findings on this topic.

In this regard, different strategies, including maximizing the initial treatment regimen, switching to another AD, or initiating a combination or augmentation therapy, were extensively utilized in order to enhance suboptimal responses to ADs [8]. However, there is a lack of agreement regarding the best option or the order that these different strategies ought to be used [9].

Despite the lack of consensus about the best treatment option, TRD is associated with social, occupational, and physical impairment, as well as high rates of mortality related to suicidal behaviors. Compared to non-resistant MDD, TRD is associated with a substantial increase in direct and indirect medical care costs, mainly due to recurrence and long hospitalizations [10]. In this sense, the identification of the most effective pharmacological strategies for TRD treatment may help the clinician prescribe an augmentative compound to AD early.

In light of these considerations, the purpose of the present review is to update the available data regarding the efficacy of different pharmacological strategies as augmentation in TRD in order to guide the clinician through the complex choice of the most adequate pharmacological treatments.

## 2. Materials and Methods

A search in the main psychiatric databases was performed (PubMed, ISI Web of Knowledge, PsychInfo). Original articles in English language from 1956 to 30 April 2020, with available abstract and full texts, were included.

The search was performed using the keywords “augmentation major depression”. At least two authors, who subsequently checked and extracted data from included articles, carried out the selection of appropriate papers. The first selection was performed considering the pertinence of the title and the information given in the abstract; the second selection was made after accurate reading of the research methods in the full-text.

Inclusion criteria were: (1) original articles; (2) mean age of patients over 18 years; (3) reported diagnosis of unipolar major depressive disorder [11] resistant to AD treatments; (4) clear definition of the treatment resistance using criteria widely accepted in literature (no response to one or two AD trials at adequate dosages and for adequate period of time; reduction of psychometric scores ≤ 50%); (5) mood improvement as primary outcome; (6) topic of the article focused on pharmacological augmentation in TRD.

Exclusion criteria were: (1) reviews, meta-analyses, commentaries, letters, case reports, pooled analyses, comments, case studies, study protocols; (2) mean age of the subjects under 18 years; (3) studies with combined treatments (augmentation is defined as the addition of a new compound to previous antidepressant); (4) mixed or not accurately described diagnoses (e.g., mixed group with bipolar and unipolar depression, anxiety), unless data were available for the subgroup of unipolar patients; (5) not accurately defined treatment resistance; (6) studies conducted on animals; (7) non-pharmacological augmentation (e.g., psychotherapies or neuromodulation techniques); (8) articles written not in English language. This section may be divided by subheadings.

## 3. Results

The search of all the databases provided a total of 1499 citations. Among these, 44 were identified as duplicates. Four-hundred-ninety studies were discarded because, after reviewing the abstracts, the papers dealt with another topic, and 52 studies were discarded because they were written in a different language. Reviews and meta-analyses (*n* = 433), as well as comments, case reports, and letters (*n* = 94), were excluded. Moreover, 31 papers were discarded because the studies were conducted on animals and 13 because a paediatric population was examined.

The full texts of the remaining 342 citations were examined in more detail, concluding that 235 studies did not meet the inclusion criteria because: one-hundred-twenty-four did not accurately define treatment resistance; forty-nine studies included mixed samples; forty-one articles were about neuromodulation; thirteen were pooled analyses; four were study protocols; four were duplicates.

One-hundred-seven studies met the inclusion criteria and were included in the present review (Figure 1).

### 3.1. Other Antidepressants and Buspirone

The pharmacological augmentation with other ADs, such as selective serotonin reuptake inhibitors (SSRIs), serotonin and norepinephrine reuptake inhibitors (SNRIs), or tricyclic antidepressants (TCA), in TRD has been poorly investigated until now, additionally in light of the synergic effect and the concrete risk of adverse events [12]. Altamura and his group [13] published a randomized placebo-controlled study (RCT) demonstrating the efficacy of augmentative low-dose intravenous (IV) citalopram to oral SSRIs in 36 non-responders affected by MDD. Similarly, an open-label study reported the effectiveness of buspirone augmentation in patients with a poor response to SSRIs [14], and Taylor and Prather [15] demonstrated the effectiveness of augmentative nefazodone to ADs in a small group of 11 patients with TRD and presenting high levels of anxiety.

On the other hand, in a large double-blind RCT conducted by Licht and Qvitzau [16], the authors demonstrated that sertraline monotherapy at 100 mg/day has similar effects to a mianserin add-on, and, in a more recent RCT, mirtazapine did not show superiority over a placebo in improving the depressive symptoms in a large sample of TRD patients (*n* = 431) [17]. A description of the cited studies is reported in Table 1.

#### Head-to-Head Studies

It has been demonstrated that short-term augmentation with low-dose IV citalopram or clomipramine is more effective than augmentation with a placebo in 54 patients with no response to SSRIs [18].

Three studies compared the efficacy of continuation with AD monotherapy versus augmentation with TCA/SSRIs or lithium. Fava and collaborators [19,20] published two double-blind studies including fluoxetine-resistant MDD individuals who were randomly assigned to higher dosages of fluoxetine, fluoxetine plus desipramine, or fluoxetine plus lithium. In the first study, including 41 subjects, high-dose fluoxetine was more efficacious than augmentative treatments. In contrast, the subsequent study (*n* = 101) showed no significant differences in the response rates across the three treatment groups. Finally, a recent randomized open-label study showed a higher percentage of response in favor of add-on citalopram with respect to add-on lithium in 104 MDD individuals treated unsuccessfully for 10 weeks with imipramine [21] (Table 1).

### 3.2. Second Generation Antipsychotics (SGAs)

#### 3.2.1. Aripiprazole

Aripiprazole was the most investigated molecule among SGAs. A first retrospective chart review [22] showed the effectiveness of augmentative aripiprazole in 30 TRD patients. A more recent retrospective study reported significant improvements in both the Montgomery and Åsberg Depression Rating Scale (MADRS) and Young Mania Rating Scale (YMRS) total scores in 38 TRD patients, presenting a mixed specifier according to the Diagnostic and Statistical Manual of Mental Disorders—5th edition (DSM-5) [23].

Eight open-label studies reported positive results. The authors of two papers [24,25], involving a maximum of 15 TRD patients, demonstrated the effectiveness of aripiprazole in augmentation to AD treatment. These results were confirmed in the long-term in a large sample [26]. Fabrazzo and colleagues [27] demonstrated that a fixed dose of aripiprazole (5 mg/day) was effective in augmentation to TCA in 35 TRD patients. Adjunctive aripiprazole was equally effective when added to paroxetine or sertraline in 24 MDD patients [28], and low-dose augmentative aripiprazole also demonstrated its effectiveness in 9 TRD Taiwanese patients [29]. A 12-week prospective open-label multicenter study reported a significant amelioration of depressive symptoms with augmentative aripiprazole as shown by an endpoint response rate higher than 50% and a remission rate of about 40% [30]. Recently, Horikoshi and his group [31] published a randomized open-label study demonstrating the effectiveness of both low-dose (LD; 3 mg/day) and high-dose (HD; 12 mg/day) aripiprazole augmentation in TRD, with a statistically significant difference in terms of earliness of response in the HD versus LD group.

Three multicenter RCTs, conducted by the same research group [32,33,34], demonstrated the efficacy of the molecule in large samples (*n* = 362; *n* = 324; *n* = 349, respectively) of TRD patients. In contrast, an RCT [35] failed to find the superiority of augmentative low-dose aripiprazole (2–5 mg/day) versus a placebo. This latter sample (*n* = 221) was re-analyzed and the results were reported in two other papers. The first one [36] showed that a modest percentage of non-responders to 2 mg/day ameliorated with aripiprazole 5 mg/day, while the second [37] demonstrated a statistically significant improvement on the depression subscale of the Kellner Symptom Questionnaire but without statistically significant effects on the anxiety and hostility subscales. A further RCT was conducted on 181 patients suffering from late-life depression, and it demonstrated the efficacy of a maximum 15 mg/day augmentative dose of aripiprazole [38]. Finally, a Japanese research group carried out the aripiprazole depression multicenter efficacy (ADMIRE) study randomizing 540 patients to a placebo, fixed-dose (3 mg/d), or flexible-dose (3–15 mg/d) of aripiprazole. The groups with active treatment showed more improvement in depressive symptoms than the placebo [39], as also confirmed by a sub-analysis on the *core* depressive symptoms [40].

#### 3.2.2. Other SGAs

Promising results were reported for brexpiprazole, a molecule with some pharmacological similarities to aripiprazole. Two RCTs conducted on large samples by Thase and his group [41,42] demonstrated the superiority of brexpiprazole over a placebo at the highest dosages of 2 or 3 mg/day, but not at 1 mg/day. More recently, an open-label study with 51 TRD patients who had not responded to previous augmentative treatment demonstrated a significant reduction in the MADRS scores with brexpiprazole augmentation independently from the prior augmentative strategy [43]. One double-blind RCT confirmed that brexpiprazole was better than a placebo in reducing the MADRS scores among 393 TRD patients [44].

The results from a three-phase study, conducted on 489 patients, demonstrated that augmentative risperidone ameliorated depressive symptoms in the short period, but it was not more efficacious than a placebo in the long-term [45]. A sub-analysis of this latter sample showed the significant amelioration of depression with augmentative risperidone in 89 elderly patients who had not responded to citalopram [46]. A further RCT by Mahmoud and colleagues [47] reported the statistically significant improvement in the depressive symptoms in patients treated with risperidone compared to a placebo.

In one open-label study conducted on 20 TRD patients, Papakostas and his group [48] demonstrated the effectiveness of ziprasidone in augmentation to SSRIs.

Two open-label small sample studies suggested beneficial effects of quetiapine as augmentation treatment in TRD [49,50].

The augmentation of fluoxetine with olanzapine showed to be more efficacious than olanzapine or fluoxetine monotherapy in a double-blind small sample RCT [51]. These results were similar to those reported in an open-label study assessing the effectiveness of augmentative olanzapine in 11 TRD patients treated with milnacipran [52].

Finally, a recent RCT (*n* = 530) failed to prove the efficacy of augmentation with cariprazine in patients with MDD and inadequate previous response to antidepressants [53]. 

A description of the studies regarding SGA augmentation in TRD is reported in Table 2.

#### 3.2.3. Head-to-Head Studies

No significant differences in terms of effectiveness were found between quetiapine extended release (XR) at 300 mg/day and lithium as add-on therapies [54] in 557 TRD patients. Furthermore, brexpiprazole, but not quetiapine, resulted to be superior to a placebo in a further double-blind RCT study [55].

Gobbi and collaborators [56] reported that both the augmentation with another AD or with a SGA (olanzapine, risperidone, quetiapine, aripiprazole) improved depressive symptoms over time; hovewer, SGAs performed better than Ads (Table 2).

#### 3.2.4. Switch versus Augmentation

In a randomized study by Mohamed and co-authors [57], aripiprazole augmentation was significantly more effective than switching to bupropion monotherapy in terms of remission rates, while the difference regarding the augmentation strategy for both the compounds did not result to be statistically significant (Table 2).

### 3.3. Mood Stabilizers

#### 3.3.1. Lithium

Among the treatment strategies for patients with TRD, the add-on of lithium to an ongoing AD therapy is one of the most widely studied treatment options.

A first double-blind RCT with 34 tricyclic-resistant depressed patients [58] reported that higher range doses of lithium (750 mg/day), but not low ones (250 mg/day), may have a significant AD effect. In contrast, a subsequent study on 92 TRD subjects found no significant difference between lithium and placebo augmentation in 35 subjects resistant to nortriptyline treatment [59].

One RCT, evaluating the efficacy of lithium augmentation in the 4-month continuation treatment of 27 TRD patients, showed that relapses occurred only in the placebo group, thus suggesting that patients responding to lithium augmentation should continue the treatment for a minimum of 6 months or even longer [60]. Two years later, Bschor and collaborators [61] studied 22 individuals from the same sample, finding no significant differences between lithium and placebo in terms of 1-year recurrence rates and concluding that lithium should be maintained for at least 1 year after reaching response to lithium augmentation in TRD patients.

Most of the open-label studies reported promising results for lithium augmentation in TRD subjects. Thase and collaborators [62] demonstrated its effectiveness in 20 non-responders to imipramine and psychotherapy. A subsequent open-label study reported a significant improvement in 13 out of 20 patients suffering from MDD resistant to desipramine [63], and, in the same year, Fontaine and colleagues [64] showed the effectiveness of lithium augmentation in 60 outpatients with MDD resistant to desipramine or fluoxetine. Subsequently, an open-label dose-response study with 11 sertraline-resistant MDD subjects demonstrated that most patients responded within 1 week, but the degree of response was not associated with lithium plasma levels [65]. Some years later, an open-label study with 22 depressed outpatients not responding to venlafaxine showed that eight of them responded and two reached remission [66]. Similarly, another open-label study, carried out on 13 MDD patients not responding to venlafaxine, reported that 38.5% of the total sample achieved a response with lithium augmentation [67].

Two open-label studies [68,69] failed to find a significant improvement in the depressive symptoms of, respectively, 13 and 21 elderly TRD patients. In contrast, a recent multicenter cohort study reported a greater clinical amelioration with lithium add-on therapy in geriatric TRD patients versus non-geriatric ones [70].

A description of the studies regarding lithium augmentation in TRD is reported in Table 3.

#### 3.3.2. Head-to-Head Studies: Lithium

One open trial, including 28 elderly TRD inpatients, showed that the response and remission rates were higher in the lithium augmentation group than in the phenelzine one [71].

Two studies compared lithium with SGA as an add-on therapy in TRD. The first open-label study randomized 20 TRD patients to either lithium or quetiapine add-on. The scores of the Hamilton Rating Scale for Depression (HAM-D) significantly improved from the baseline in both groups, with a greater improvement in the quetiapine than the lithium group after the first 4 weeks of augmentation [72]. The second study was an RCT conducted on a sample of 30 TRD patients who did not show a significant difference in clinical amelioration with lithium, olanzapine, or aripiprazole augmentation [73].

Three studies compared lithium versus anticonvulsant augmentation.

In the first randomized, open-label study, including 34 TRD patients, no significant differences were observed in the HAM-D scores between lamotrigine and lithium treatment groups, both at baseline and after 8 weeks [74]. In contrast, an open-label trial, including 88 TRD patients, reported significant clinical improvement within the second week in the lamotrigine group compared to the lithium group, while no differences were observed at the study endpoint [75]. The third open-label study, conducted on 46 TRD inpatients, found that lithium, but not carbamazepine, significantly augmented the AD effect of mirtazapine [76].

Finally, two studies investigated the efficacy of lithium compared to triiodothyronine (T_3_) augmentation in TRD. Joffe and collaborators [77] carried out an RCT study including 50 depressed subjects resistant to TCA treatment, and they found that both T_3_ and lithium were similarly more effective than a placebo in decreasing HAM-D scores. In a subsequent observational study, lithium-sulfate augmentation was found to lead to higher remission rates than lithium-sulfate plus T_3_ augmentation [78] (Table 3).

#### 3.3.3. Lithium Augmentation versus Switching Strategies

Two studies with elderly subjects compared the effect of augmentation versus switch strategies. In the first one (*n* = 65), the authors observed similar rates and speed of response with bupropion, nortriptyline, or lithium augmentation compared to switching to venlafaxine [79]. In the second one (*n* = 32), both lithium augmentation and switching to TCA were reported to be effective strategies, differently from switching to phenelzine or to electro-convulsive therapy (ECT) [80] (Table 3).

#### 3.3.4. Antiepileptic Drugs

In a first double-blind RCT, conducted on 23 TRD patients, lamotrigine failed to significantly reduce both HAM-D and MADRS scores with respect to a placebo [81]. Similarly, another double-blind RCT showed no differences between lamotrigine and placebo treatment groups of 34 TRD subjects [82]. On the other hand, three retrospective studies reported some possible benefits of lamotrigine augmentation to an AD regimen in TRD individuals. In the first one, by Barbee and Jamhour [83], 48.4% of the 31 included patients were rated much or very much improved on the Clinical Global Impression (CGI) scale after 6 weeks of treatment. No differences were found in the doses of lamotrigine administered to responders and non-responders. A subsequent paper reported an improvement in 76% of the 25 TRD enrolled patients [84]. Finally, 34 TRD subjects treated with augmentative lamotrigine showed an early and statistically significant improvement of some target symptoms, such as depressed mood, loss of interest, cognitive impairment, irritability, and anergy, while sleep disturbance did not ameliorate [85].

A small double-blind RCT (*n* = 20) found no differences in the overall response rates between phenytoin augmentation and a placebo in MDD subjects resistant to SSRIs [86].

One open-label study, conducted on 20 elderly subjects suffering from TRD and generalized anxiety disorder, demonstrated a statistically significant reduction in depression scores after 4-week pregabalin augmentation, with a further improvement between the 8th and the 12th week; also, a significant improvement of anxiety was noted [87]. Similarly, another study conducted with an open-label design on 14 TRD subjects demonstrated the effectiveness of valproate as an augmentation treatment to ADs, lasting 7 months [88].

As some studies reported the beneficial effects of topiramate on the depressive phase of bipolar disorder, the efficacy of this drug as an add-on therapy in TRD was assessed. Mowla and Kardeh [89], in a double-blind RCT (*n* = 53), demonstrated the superiority of topiramate over a placebo in reducing HAM-D total scores; in particular, depressed mood, insomnia, agitation, anxiety symptoms, and suicidality significantly improved in the topiramate group.

Finally, zonisamide was evaluated as a possible augmentation strategy in the treatment of TRD subjects, in consideration of partial pharmacodynamic overlap with lamotrigine, as well as the possible effect on the serotonergic system; the authors concluded that zonisamide might be a potential augmentation option for MDD patients not responding to duloxetine [90].

A description of the studies regarding augmentation with antiepileptic drugs in TRD is reported in Table 4.

#### 3.3.5. Head-To-Head Studies: Antiepileptics

One retrospective study compared different molecules, including valproate, as an add-on therapy to paroxetine in 225 TRD patients. The remission rates were 48.7% for valproate, 26.7% for risperidone, 32.6% for buspirone, 42.6% for trazodone, and 37.5% for thyroid hormone, but the difference between the adjunctive treatments did not result to be statistically significant [91] (Table 4).

### 3.4. Ketamine and Es-Ketamine

Ketamine is a non-competitive N-methyl-D-aspartate (NMDA) glutamate receptor antagonist used as an anesthetic in clinical practice. Two open-label small-sample studies [92,93] reported a significant reduction in depressive symptoms when intravenous ketamine was administered as an add-on therapy.

Five double-blind RCTs presented contrasting results. Fava and colleagues [94] demonstrated that 0.5 and 1.0 mg/kg of IV ketamine, but not 0.1 and 0.2 mg/kg, were better than a placebo in reducing the HAM-D6 total score in 86 TRD subjects 24 h after the infusion, not at day 3. In contrast, Ionescu and colleagues [95] enrolled 26 TRD outpatients, showing that IV ketamine at 0.5 mg/kg did not outperform a placebo in terms of short- or long-term efficacy. Some authors reanalyzed the sample enrolled by Fava and his group [94], searching for more specific differences in terms of gender, time to relapse, or suicidal ideation. The analyses on varying ketamine doses, administered IV, on 50 men and 49 women with TRD, did not lead to significant differences between the sexes in treatment response [96]. More recently, Salloum and colleagues [97] reanalyzed a subgroup of 56 patients and demonstrated that time to relapse after varying doses (0.1-0.5-1.0 mg/kg) of a single administration of IV ketamine was dose-related. Moreover, a single infusion of IV ketamine significantly reduced suicidal ideation after one month, but not early after the administration in a subgroup of 56 TRD patients [98].

The nasal-spray formulation of esketamine, the S-enantiomer of ketamine, has been recently approved by the Food and Drug Administration (FDA) and the European Medicines Agency (EMA) as augmentation treatment in TRD. Two double-blind RCTs regarding the efficacy of esketamine in the short-term [99,100] and one regarding time to relapse in the long-term [101] supported the approval of this compound for TRD. In particular, the short-term studies reported the superiority of esketamine over a placebo as augmentation in TRD patients over a 4-week period, although this difference was not statistically significant when esketamine was administered at 84 mg twice weekly in the study by Fedgchin et al. [99]. The RTC by Daly and colleagues [101] demonstrated the superiority of esketamine over a placebo in preventing the relapse of 297 TRD patients who achieved a stable response or remission after 16 weeks of esketamine add-on therapy to their current AD treatment.

More recently, Ochs-Ross and collaborators [102] reported no differences between esketamine and a placebo in reducing the MADRS scores in a sample of elderly TRD patients, although there was a significant reduction in 65–74 years-old patients, different from subjects with an age ≥ 75 years. Promising results were also found by Wajs and colleagues [103], who reported that improvements in depressive symptoms after esketamine augmentation remained in the long-term.

A description of the studies regarding ketamine and esketamine augmentation in TRD is reported in Table 5.

### 3.5. Psychostimulants

#### 3.5.1. Flenfluramine

Flenfluramine, a sympathomimetic amine that can activate the serotoninergic pathways in the brain, was studied by Price and co-authors [104] in 15 TRD patients, demonstrating no statistically significant evidence of either transient or maintained clinical improvement during the 2 weeks of fenfluramine augmentation to desipramine (Table 5).

#### 3.5.2. Lisdexamfetamine Dimesylate

In the research paper by Richards and colleagues [105], the authors reported the results of two multicenter double-blind RCTs for a total of 826 TRD patients, showing that lisdexamfetamine dimesylate did not provide benefits over a placebo (Table 5).

#### 3.5.3. Methylphenidate

Methylphenidate (MPH), a central nervous system (CNS) stimulant, was studied as a potential augmentation therapy for patients with refractory depression, but the preliminary findings appear unpromising.

A first double-blind RCT [106] showed no statistically significant differences between the extended-release MPH and placebo in 50 TRD patients. Similarly, another double-blind RCT [107] reported no statistically significant differences between osmotic-release oral system MPH and placebo at endpoint on the MADRS total scores, although MPH was superior to the placebo in improving apathy and fatigue as measured by the multidimensional assessment of fatigue (MAF) scale and the apathy evaluation scale (AES) (Table 5).

#### 3.5.4. Modafinil

A retrospective chart review conducted by Nasr [108] on 78 TRD outpatients reported preliminary but promising findings about augmentation with modafinil, a wake-promoting agent. In particular, 11 depressed subjects achieved remission and most patients improved in depressive symptoms, particularly sleepiness, fatigue, and anergy (Table 5).

### 3.6. Non-Psychopharmacological Agents

#### 3.6.1. Acetylsalicylic Acid (ASA)

Mendlewicz and colleagues [109] reported that ASA augmentation to SSRIs in 17 TRD patients significantly reduced the HAM-D total scores, and it was associated with a response rate of 52.4% and remission rate of 43%. Moreover, 82% of responders achieved remission (Table 6).

#### 3.6.2. Metyrapone

Metyrapone, a blocker of cortisol synthesis, was investigated in a double-blind RCT on 165 patients, showing no significant amelioration of depressive symptoms compared to a placebo [110] (Table 6).

#### 3.6.3. Minocycline

One open pilot trial conducted by Avari and colleagues [111] on a small sample of older adults affected by TRD reported that, after 8 weeks of augmentation with minocycline, the remitters represented 31% of the sample. A year later, one RCT was conducted on 39 TRD subjects who took minocycline (200 mg/day) as an adjunctive treatment to the AD one. The authors did not demonstrate a significant superiority of minocycline with respect to a placebo in improving the HAM-D scores at week 4 [112]. The difference became significant when the patients were stratified for the baseline C-reactive protein (CRP) plasma levels, showing a greater improvement in the CRP+ group (CRP ≥ 3 mg/L) treated with minocycline than all the other groups (Table 6).

#### 3.6.4. Pindolol

Two double-blind RCTs were conducted on small samples of TRD patients (*n* = 10 and *n* = 9, respectively). In the first one, by Moreno and co-authors [113], pindolol did not show a statistically significant difference with respect to a placebo at the endpoint in terms of the amelioration of depressive symptoms. In contrast, in the second one, the patients exhibited significant improvement on HAM-D total scores; the authors argued that a single high dose of pindolol (7.5 mg) is a more effective augmentation strategy in SSRI-refractory patients compared with the same total dose given at 2.5 mg t.i.d. [114] (Table 6).

Two further double-blind RCTs on larger samples failed to demonstrate the efficacy of pindolol as an augmentative agent in TRD. Perez and colleagues [115] enrolled 80 TRD outpatients and showed that the HAM-D scores and remission rates were not significantly different in patients taking a placebo or pindolol (2.5 mg t.i.d.). Moreover, Perry and collaborators [116], using a hemi-crossover design, showed that there were no significant differences in AD response between the subjects receiving a placebo or pindolol as add-on therapy to SSRI monotherapy (Table 6).

#### 3.6.5. Reserpine

Price and colleagues [117] reported the results of eight patients affected by melancholic depression and who were treated with reserpine in augmentation to desipramine. Only one patient had a resolution of the depressive and psychotic symptoms within 48 h, but he relapsed within 2 weeks; the depressive symptoms did not significantly ameliorate in the total sample (Table 6).

#### 3.6.6. Testosterone

Miller and collaborators [118] tested low-dose transdermal testosterone in a group of nine women with TRD using an open-label pilot protocol. The authors reported a statistically significant improvement in the mean MADRS scores beginning from week 2 and maintained during the 8-week follow-up period. Two-thirds of the subjects achieved a response and one-third remitted. Differently, a double-blind RTC recently conducted in a larger sample of women (*n* = 101) did not demonstrate a significant superiority of low-dose testosterone to a placebo after 8 weeks of augmentation in improving depressive symptoms, fatigue, and sexual function [119] (Table 6).

#### 3.6.7. T3/T4

Rudas and his group [120] conducted an open study with nine TRD patients, showing a significant reduction in the HAM-D total scores; in particular, four of them reached a good response to high-dose T4 (Table 6).

### 3.7. Other Molecules and Supplements

#### 3.7.1. Anti-Parkinson/Dementia Agents

One open trial reported a significant reduction in the psychometric scores in 12 TRD subjects receiving pramipexole as an augmentative treatment [121]. Similarly, Cassano and his group [122], adding ropinirole to TCA or SSRIs in a prospective open trial, obtained a significant decrease in the MADRS total score at the endpoint in seven TRD patients (Table 6).

#### 3.7.2. Naltrexone

In a pilot double-blind RCT of low-dose naltrexone (LDN) (1 mg b.i.d.) versus placebo augmentation conducted on 12 TRD patients, the authors documented that neither the main outcome (HAM-D scores) or global outcome (CGI scores) measures presented a significant improvement for the LDN group over a placebo, with only the MADRS scores (secondary outcome) attaining a significant difference between the LDN and placebo (Mischoulon et al. 2017) [123].

#### 3.7.3. S-adenosyl Methionine (SAMe)

De Berardis and his group [124] conducted an open-label, single-blind study: 33 patients with stage II TRD, according to Thase and Rush classification [6], received SAMe in addition to the existing treatment, showing a significant reduction in the HAM-D total scores. In contrast, in a double-blind RCT carried out by Targum and collaborators [125] on 234 TRD subjects, no statistically significant difference was found between the SAMe and placebo groups (Table 6).

#### 3.7.4. Supplements

Siwek and co-authors [126], in a double-blind RCT, investigated the effect of adjunctive zinc in 21 patients resistant to imipramine, demonstrating a significant decrease in the depression scores of zinc in comparison to a placebo (Table 6).

## 4. Discussion and Expert Opinion

Despite the fact that the evidence regarding the comparative effectiveness of the common treatments for MDD is evolving [127], to date, the heterogeneous characteristics of depression complicate the treatment options. Particularly, TRD is associated with a failure to respond to different treatment trials, reflecting the complexity of this multifactorial disorder. However, several strategies have been proposed, such as dose optimization, the switch to another therapeutic class, or augmentation. Although augmentation allows the limitation of the transition period between one antidepressant to another, the evidence comparing augmentation versus other strategies is limited, sparse, and equivocal. Moreover, previous meta-analyses have not generated clear conclusions with regard to the efficacy of the available augmentation agents since they have been restricted by small sample size and a paucity of direct comparators between the treatments [128,129]. Nevertheless, the poly-therapy related to augmentation strategies may lead to side effect burden and complicate adherence. However, some evidence-based considerations should be made about the effectiveness of augmentation agents in adult TRD patients.

According to the present review, lithium and aripiprazole represent the most studied agents in relation to this topic. Lithium add-on is one of the oldest and most established augmentation strategies. At present, augmentative lithium did not show more efficacy than a placebo in three RCTs for a total of 91 patients both in acute treatment [58,59] and in prevention of relapses [61], with the exception of one 4-month follow-up study [60]. On the other hand, eight independent open-label studies, for a total of 334 patients, reported promising findings regarding augmentation with lithium in TRD patients [62,63,64,65,66,68], including elderly subjects [69,70]. Only one small-sample open-label trial did not demonstrate the superiority of lithium over a placebo [67]. Furthermore, the head-to-head studies predominantly confirmed that lithium add-on was as effective as adjunctive antidepressants, SGAs, antiepileptics, and T3. Taken as a whole, the studies on lithium suggest a potential role of this compound as an augmentative treatment in TRD. However, most of these studies involved augmenting tricyclic antidepressants. Therefore, evidence on the use of lithium as an add-on therapy to current antidepressants (e.g., SSRIs and SNRIs) is limited. In addition, whilst lithium augmentation may decrease suicide risk [130], the multiple drug–drug interactions, together with its risks of toxicity, require the frequent monitoring of the lithium serum concentration.

Other augmentation strategies with the most evidence-based support include atypical antipsychotics and, more recently, ketamine and esketamine. While the exact mechanism of augmentation is not entirely known, it is hypothesized that SGAs can act as 5-HT2A receptor antagonists, alpha 2 adrenergic antagonists, 5-HT1A agonists, and monoamine reuptake inhibitors [131].

Among the SGAs, aripiprazole has been the most studied agent. According to our review, we included 19 studies with nine RCTs involving a total of 1977 patients [32,33,34,35,36,37,38,39,40]. All the RCTs, with the exception of one [35], showed the superiority of aripiprazole over a placebo in improving the depressive symptoms of the TRD subjects, both at fixed and flexible medium-low doses [39]. The effectiveness of the augmentation with aripiprazole was also confirmed by open-label trials for a total of 1253 patients [22,23,24,25,26,27,28,29,30,31]. One of these studies reported that medium doses (12 mg/day) were more effective than low doses (3 mg/day) [31]. Moreover, olanzapine and quetiapine showed promising results, although drawn from studies with very limited samples [49,50,51,52]. To date, the FDA has approved aripiprazole, as well as quetiapine and olanzapine plus fluoxetine, as therapeutic options for major depression or specifically for TRD. Although risperidone has not been FDA-approved for the treatment of TRD, it should be used clinically as an augmentation strategy. Indeed, risperidone showed superiority in ameliorating acute symptoms [47] and preventing relapses compared to a placebo in three RCTs for a total of 616 subjects [45,46]. On the other hand, the data concerning ziprasidone are both limited and mixed [48]. Apart from its relatively favorable side effect profile, it is not presently recommended as an augmentation strategy. Finally, the serotonin–dopamine activity modulator—brexpiprazole—is the first “third generation” antipsychotic receiving FDA approval for augmentation in TRD. So far, evidence about its efficacy is limited to the clinical trials that led to this approval [43,44]. Thus, real-world data about the efficacy and safety of brexpiprazole are particularly urged.

Evidence of glutamatercic system impairments in MDD has emerged from clinical studies reporting that the sub-anesthetic (0.5 mg/kg) IV administration of ketamine is related to a rapid antidepressant response and reduction of suicidal ideation [132]. The results regarding IV ketamine augmentation in TRD seem promising. According to our review in which three out of the five studies are RCTs, including a total of 112 patients [94,95,97], it has been reported that low doses of the molecule have a rapid and significant AD effect. However, no studies indicate that the antidepressant acute effects of ketamine are maintained in the long-term. Nevertheless, adverse events have been reported in different studies, and the IV formulation of ketamine may hamper its administration in several clinical settings [133]. In this framework, esketamine, the S-enantiomer of ketamine, has about a four-fold greater affinity for the glutamate receptor than ketamine, thus allowing the use of much lower doses and reducing the risk of dose-dependent dissociative symptoms associated with ketamine administration. Although the nasal-spray formulation of esketamine was approved by the FDA after showing its efficacy in TRD patients, the clinical relevance of esketamine use after the induction phase is not fully understood. Although many studies focused on the short-term efficacy of esketamine [99,100,102], namely the 4-week induction phase, our knowledge about the long-term effect of this compound is poor, both in patients who discontinued the treatment after the induction phase and those who continued the AD. Continued esketamine treatment following the induction phase may be associated with stable efficacy in relapse prevention among TRD patients [101,103]. However, the long-term antidepressant and anti-suicidal effects of esketamine after discontinuation might be inconsistent [134]. Indeed, TRD has a chronic course, poor clinical stabilization, and high suicidal risk, and, therefore, to date, it is difficult to know whether clinicians should continue to use esketamine after the acute episode, for how long, at which dose, and in which patients. Moreover, another relevant question is how esketamine should be discontinued. The potential abuse and effects on cognition should be taken into account, especially in the case of prolonged administration [135,136]. Therefore, other controlled studies, more adherent to the real-world clinical practice, are needed to evaluate their efficacy and safety in the long-term, also in light of the potential risk of abuse of intranasal esketamine [127,128]. Moreover, to date, this molecule cannot be administered at home, thus limiting an easy use in clinical practice.

Finally, the augmentation of an AD with another one has limited evidence, and add-on therapy with antiepileptic drugs, pramipexole, ropinirole ASA, metyrapone, reserpine, testosterone, T3/T4, naltrexone, SAMe, modafinil, amphetamines, or zinc cannot be recommended on the basis of the current evidence.

Despite the limitations of this review, including the heterogeneity of the included studies in terms of the duration and doses of the augmentative treatment, setting of care (inpatients/outpatients), class of augmented AD and design, the focus on efficacy and not on safety and tolerability data, as well as the strict and specific inclusion criteria that may have limited the number of eligible studies, we suggest that future studies should take into account the following observations: –use of appropriate TRD definition, more homogeneous populations, as well as rating scales for the evaluation of severity of depression in order to allow pooled analyses and meta-analytical approaches of large samples of patients [129];–more clarity in defining whether certain symptoms are part of depression or the result of comorbid psychiatric conditions [137];–larger samples are required to obtain more reliable data;–combined treatments might be associated with potential dangerous side effects, so future studies should quantify the risk/benefit ratio of different augmentation strategies;–pharmacokinetic interactions should be monitored in the long-term, measuring drug plasma levels regularly.–Furthermore, the directions of future research regarding augmentation agents for TRD treatment could be the following:–understanding the biological nature of treatment response, thus increasing biological insights into the pathophysiology of TRD;–focus on searching for other molecules with more potent and longer-lasting antidepressant effects than esketamine;–identification of more robust biomarkers for clinical practice, allowing early interventions and, ultimately, improving remission outcomes.

## 5. Conclusions

Our search identified different pharmacological augmentation treatment options for TRD. Taken as a whole, the common prescription of SGAs as an add-on therapy to AD in TRD patients seems to be supported by some evidence of efficacy, in particular for aripiprazole. Among the other molecules, lithium might represent a valid therapeutic option, but more studies are needed to draw definitive conclusions and, given the relatively high burden of potential acute and chronic side effects, lithium augmentation should be considered a second-line choice. Esketamine has shown some data of efficacy in TRD patients, but more evidence from the real world of clinical practice is needed to implement its use in the long-term. Overall, clinicians should interpret these findings cautiously in light of the evidence of potential treatment-related adverse effects.

## Figures and Tables

**Figure 1 ijms-22-13070-f001:**
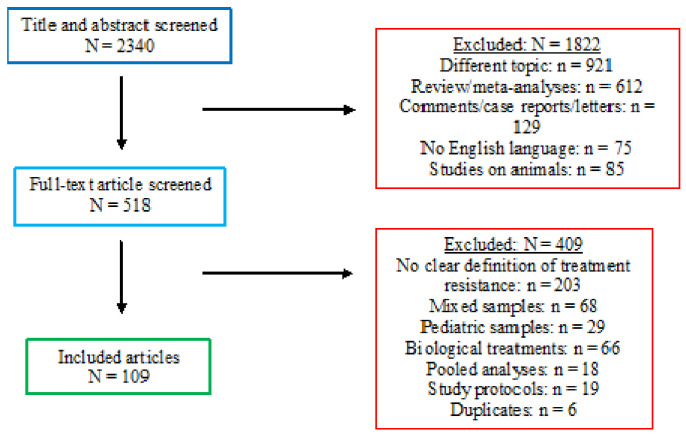
PRISMA flow-chart for systematic reviews.

**Table 1 ijms-22-13070-t001:** Summary of the studies about augmentation with other ADs and buspirone in TRD.

Reference	*n*	Age (Years)	Design	Augmentation Molecule	Dosage	AD	Duration	Primary Outcome Measures	Results
**RCTs**									
Altamura et al. [13]	22	18–65	Single-blind Comparison IV CIT/pcb	Citalopram	10 mg/d (IV)	Paroxetine, Sertraline or Escitalopram	5 days	HAM-D, MADRS	↓ HAM-D: CIT > pcb (*p* < 0.01) ↓ MADRS: CIT > pcb (*p* < 0.05)
Licht and Qvitzau [16]	253	Mean (SD): 39 (±12)	Multicenter, double-blind Comparison SER100 + pcb/SER200 + pcb/SER100 + MIA	Mianserin	30 mg/d	Sertraline	5 weeks	HAM-D	↓ HAM-D: - SER100 = SER100 + MIA (*p* = 0.85) - Both > SER200 (*p* < 0.05)
Kessler et al. [17]	431	18–65	Multicenter, double-blind Comparison mirtazapine/pcb	Mirtazapine	30 mg/d	SSRIs or SNRIs	1 year	BDI-II at week 12	↓ BDI-II: mirtazapine = pcb (*p* = 0.09)
Altamura et al. [18]	54	18–65	Head-to-head Single-blind Comparison IV CIT/CLO/pcb	Citalopram/clomipramine	CIT: 10 mg/d CLO: 25 mg/d	SSRIs	5 days	HAM-D melancholy and anxiety-somatization scores	↓ HAM-D: - CIT and CLO > pcb (*p* < 0.01) CIT = CLO on melancholy (*p* = 0.73) CIT > CLO on anxiety/somatization (*p* = 0.027)
Fava et al. [19]	41	Mean (SD): 39.6 (±9.9)	Head-to-head Double-blind Comparison FLU/FLU + Li/FLU + DES	Desipramine	FLU: 40–60 mg/d FLU 20 mg/d + Li: 300–600 mg/d FLU 20 mg/d + DES: 25–50 mg/d	Fluoxetine	4 weeks	HAM-D	↓ HAM-D: Whole sample: FLU > FLU + Li/FLU + DES (*p* = 0.05) Non-responders: FLU/FLU + Li > FLU + DES (*p* = 0.04)
Fava et al. [20]	101	Mean (SD): 41.6 (±10.6)	Head-to-head Double-blind Comparison FLU/FLU + Li/FLU + DES/pcb	Desipramine	≈	Fluoxetine	4 weeks	HAM-D	FLU = FLU + Li = FLU + DES = pcb (*p* = 0.20)
Open studies									
Joffe and Schuller [14]	25	Range: 22–56 Mean: 40.2	Open-label	Buspirone	Range: 20–50 mg/d Mean: 31.2 mg/d	Fluoxetine or Fluvoxamine	3 weeks	CGI	Response rate: 68% Remission rate: 32%
Taylor and Prather [15]	11	Mean (SD): 53 (±10.5)	Open-label	Nefazodone	50–300 mg/d Mean (SD): 200 (±134.2) mg/d	Various ADs	9 months	Presence/absence anxiety/depression symptoms CGI	Depression remission (*p* < 0.005) Anxiety remission (*p* < 0.0005)
Navarro et al. [21]	104	Mean (SD): 47.9 (±8.4)	Head-to-head Open-label Comparison CIT/Li	Citalopram	CIT: 30 mg/d Li: 600 mg/d	Imipramine	10 weeks	HAM-D	Remission: CIT > Li (*p* = 0.034) ↓ HAM-D: CIT > Li (*p* = 0.005)

Key: ≈ = same as above; AD = antidepressant; BDI-II = Beck Depression Inventory—II; CGI = Clinical Global Impression; CIT = citalopram; CLO = clomipramine; d = day; DES = desipramine; FLU = fluoxetine; HAM-D = Hamilton Depression Rating Scale; IV = intravenous; Li = lithium; MADRS = Montgomery and Åsberg Depression Rating Scale; MIA = mianserin; pcb = placebo; RCT = randomized controlled trial; SD = standard deviation; SER = sertraline; SER100 = sertraline at 100 mg/d; SER200 = sertraline at 200 mg/d; SNRI = Serotonin and Norepinephrine Reuptake Inhibitor; SSRI = Selective Serotonin Reuptake Inhibitor.

**Table 2 ijms-22-13070-t002:** Summary of the studies about augmentation with SGAs in TRD.

Reference	*n*	Age Mean (SD), y	Design	Augmentation Molecule	Dosage, mg/d	AD	Duration	Primary Outcome Measures	Results
**RCTs**									
Berman et al. [32]	362	46.5 (±10.6)	Multicenter, double-blind	Aripiprazole	2–20 Mean: 11.8	Escitalopram, fluoxetine, paroxetine, sertraline, or venlafaxine	6 weeks	MADRS	↓ MADRS: ARI > pcb (*p* < 0.001)
Berman et al. [34]	349	45.4 (±10.9)	Multicenter, double-blind	Aripiprazole	2–20 Mean: 10.7	≈	6 weeks	MADRS	↓ MADRS: ARI > pcb (*p* < 0.001)
Fava et al. [35]	221	45.4 (±10.3)	Multicenter, double-blind Phase I: ARI 2 mg/d/pcb Phase II: ARI 5 mg/d/pcb/ARI 2 mg/d	Aripiprazole	2 or 5	SSRIs or SNRIs	12 weeks	MADRS	↓ MADRS: ARI = pcb (*p* = 0.06) Response rate: ARI = pcb (*p* = 0.18) Remission rate: ARI = pcb (*p* = 0.50)
Mischoulon et al. [36] *	≈	≈	≈	Aripiprazole	≈	≈	≈	MADRS	↓ MADRS: ARI 5 mg/d > ARI 2 mg/d (*p* < 0.0001)
Dording et al. [37] *	≈	≈	≈	Aripiprazole	≈	≈	≈	KSQ	↓ KSQ depressive subscale: ARI > pcb (*p* = 0.03) ↓ KSQ anxiety, somatic, anger-hostility subscales: ARI = pcb (*p* > 0.05)
Kamijima et al. “ADMIRE” [39]	540	Fixed-ARI: 39.2 (±9.1) Flexi-ARI: 38.1 (±9.6) Pcb: 38.7 (±9.2)	Multicenter, double-blind Comparison fixed-ARI/flexi-ARI/pcb	Aripiprazole	Fixed-ARI: 3 Flexi-ARI: 3–15 (mean: 9.8)	Sertraline, fluvoxamine, paroxetine, milnacipran, duloxetine	6 weeks	MADRS	↓ MADRS: fixed-ARI = flexi-ARI > pcb (*p* < 0.01) Response and remission rates: fixed-ARI = flexi-ARI > pcb (*p* < 0.05)
Ozaki et al. [40] **	≈	≈	Subgroup analysis of ADMIRE according to: sex, age, number of AD trials, MDD diagnosis, number of depressive episodes, duration of current episode, age at first episode, time since first episode, type of SSRI/SNRI, severity at the end of AD treatment	Aripiprazole	≈	≈	≈	MADRS in subgroups; MADRS and HAM-D single items	No interaction effects of treatment and subgroups; ↓ *core* depressive symptoms (*p* < 0.05)
Lenze et al. [38]	181	66 27% of the total sample > 70	Double-blind	Aripiprazole	2–15 Mean remitters: 7 Mean non-remitters: 10	Venlafaxine	12 weeks	MADRS	Remission rate: ARI > pcb (*p* = 0.03)
Marcus et al. [33]	324	44.6 (±11.0)	Multicenter, double-blind	Aripiprazole	Mean: 11	Escitalopram, fluoxetine, paroxetine, sertraline, or venlafaxine	6 weeks	MADRS	↓ MADRS: ARI > pcb (*p* = 0.001) (d = 0.35) Remission rates: ARI > pcb (*p* = 0.016) Response rates: ARI > pcb (*p* < 0.001)
Hobart et al. [44]	393	BREX: 43.0 (±12.7) pcb: 42.7 (±12.5)	Double-blind	Brexpiprazole	2	SSRIs or SNRIs	6 weeks	MADRS	↓ MADRS: BREX > pcb (*p* = 0.007)
Hobart et al.	443	BREX: 43.6 (±11.5) QUE-XR: 44.6 (±11.6) pcb: 41.8 (±11.7)	Head-to-head Multicenter, double-blind Comparison BREX/QUE-XR/pcb	Brexpiprazole Quetiapine	BREX: Range: 2–3 Mean: 2.2 QUE-XR: Range: 150–300Mean: 198.5	SSRIs or SNRIs	6 weeks	MADRS	↓ MADRS: BREX > pcb (*p* = 0.008) QUE-XR = pcb (*p* = 0.66)
Thase et al. [41]	353	BREX: 44.1 (±11.6) pcb: 45.2 (±11.3)	Multicenter, double-blind	Brexpiprazole	2	SSRIs or SNRIs	6 weeks	MADRS	↓ MADRS: BREX > pcb (*p* < 0.001)
Thase et al. [42]	627	BREX-1: 45.7 (±11.6) BREX-3: 44.5 (±11.2) pcb: 46.6 (±11.0)	Multicenter, double-blind Comparison BREX-1/BREX-3/pcb	Brexpiprazole	1 or 3	SSRIs or SNRIs	6 weeks	MADRS	↓ MADRS: BREX-3 > pcb (*p* = 0.008) BREX-1 = pcb (*p* = 0.07)
Earley et al. [53]	435	44.2 (±11.6)	Double-blind	Cariprazine	1.5–4.5 Mean: 2.97	Various ADs	8 weeks	MADRS	↓ MADRS: CARI = pcb (*p* = 0.79)
Shelton et al. [51]	28	42.0 (±11.0)	6-week open-label fluoxetine in escalating doses; 8-week, double-blind, RCT: Comparison OLA + pcb/FLU + pcb/FLU + OLA	Olanzapine	Mean (SD) OLA + pcb: 12.5 (±5.3) Mean (SD) FLU + OLA: 13.5 (±4.1)	Fluoxetine	8 weeks	MADRS, HAM-D	↓ MADRS: FLU + OLA > OLA + pcb (*p* = 0.03) and FLU + pcb (*p* = 0.006) ↓ HAM-D: FLU + OLA > OLA + pcb (*p* = 0.03) FLU + OLA = FLU + pcb (*p* = 0.07)
Mahmoud et al. [47]	268	45.9 (±10.1)	Multicenter, double-blind	Risperidone	1–2	Various ADs	6 weeks	HAM-D	↓ HAM-D: RIS > pcb (*p* < 0.001) Response rates: RIS > pcb (*p* = 0.004) Remission rates: RIS > pcb (*p* = 0.004)
Rapaport et al. [45]	Phase I: 445 Phase II: 348 Phase III: 241	46.3 (±12.6)	Phase I: 4–6 weeks open-label CIT monotherapy Phase II: 4–6 weeks open-label RIS augmentation Phase III: 24 weeks double-blind maintenance phase, comparison CIT + RIS/CIT + pcb	Risperidone	Mean (SD): Phase II: 1.1 (±0.6) Phase III: 1.2 (±0.6)	Citalopram	(see Design)	Phases II: MADRS Phase III: time to relapse	↓ MADRS (*p* < 0.001) Time to relapse: RIS > pcb (*p* = 0.05) Relapse rates: RIS < pcb (*p* = 0.05)
Alexopoulos et al. [46] ***	Phase I: 108 Phase II: 93 Phase III: 63	63.4 (±7.9)	≈	Risperidone	Mean (SD): Phase II: 0.7 (±0.3) Phase III: 0.8 (±0.3)	≈	≈	≈	↓ MADRS (*p* < 0.001) Time to relapse: RIS > pcb (*p* = 0.07)
Open studies									
Barbee et al. [22]	19	51.2 (±8.9)	Retrospective chart review	Aripiprazole	2.5–7.5	Various ADs	6 weeks	CGI	(Very) Much improved: 52.6% Mildly improved: 26.3% Unchanged: 15.8% Minimally worse: 5.3% (*p* < 0.001)
Berman et al. [26]	987	45.8 (±11.3)	Open-label	Aripiprazole	2–30 Mean:10.1	Various ADs	52 weeks	CGI severity	CGI-S = 1 (not at all ill) or 2 (borderline ill): 69.7%
Chen et al. [29]	9	38.3 (±12.2)	Open-label	Aripiprazole	Mean: 4.2	Various ADs	4 weeks	HAM-D	Response rate: 100% Remission rate 77.8%
Fabrazzo et al. [27]	35	38.8 (±11.5)	Open-label	Aripiprazole	5	SSRIs, then CLO	24 weeks	HAM-D	↓ HAM-D (*p* < 0.0001) Response rate: 91.4% Remission rate: 34.3%
Han et al. [23]	38	28.4 (±11.3)	Retrospective MDD with mixed specifier	Aripiprazole	Mean (SD): 4.0 (±0.8)	Various ADs	8 weeks	MADRS	↓ MADRS (*p* < 0.0001) Response rate: 32% Remission rate: 21%
Hellerstein et al. [25]	14	46.1 (±13.0)	Open-label	Aripiprazole	Mean (SD): 22.5 (±9.9)	Sertraline, fluoxetine, duloxetine, venlafaxine	12 weeks	HAM-D	↓ HAM-D (*p* < 0.001) Response rate: 50% Remission rate: 28.6%
Horikoshi et al. [31]	31	LD group: 38.8 (±12.8) HD group: 44.2 (±13.9)	Open-label, R Comparison LD-ARI/HD-ARI	Aripiprazole	LD: 3 HD: 12	Various ADs	6 weeks	MADRS	↓ MADRS: LD and HD (*p* < 0.001) Response rate: HD > LD (*p* = 0.015)
Jon et al. [30]	86	45.6 (±13.7)	Multicenter, prospective, open-label	Aripiprazole	Max: 15 Mean: 6.9	SSRIs or SNRIs	6 weeks	MADRS	↓ MADRS (*p* < 0.001) Response rate 52.3% Remission rate 39.8%
Patkar et al. [24]	10	44.9 (±12.2)	Prospective, open-label	Aripiprazole	10–30 Mean: 13.2	Various ADs	6 weeks	HAM-D	↓ HAM-D (*p* < 0.001) Response rate: 70% Remission rate: 30%
Yoshimura et al. [28]	24	39.0 (±12.0)	Open-label	Aripiprazole	Mean (SD) PAR group: 8.73 (±3.13) Mean (SD) SER group: 9.23 (±3.11)	Paroxetine or sertraline	4 weeks	HAM-D	↓ HAM-D (*p* < 0.0001) PAR + ARI = SER + ARI (*p* = 0.80)
Fava et al. [43]	51	45.6 (±12.4)	Multicenter, open-label	Brexpiprazole	2.25 (±0.74)	SSRIs or SNRIs	6 weeks	MADRS	↓ MADRS (*p* < 0.0001)
Boku et al. [52]	7	53.2 (±24.0)	Open-label	Olanzapine	Mean (SD): 5.0 (±1.9)	Milnacipran	8 weeks	HAM-D, CGI	↓ HAM-D (*p* < 0.01) Response rate: 100% Remission rate: 100% ↓ CGI (*p* < 0.01)
Anderson et al. [50]	18	46.3	Open-label	Quetiapine	Max 300 mg twice Mean (SD): 245.0 (±68.0)	Various ADs	8 weeks	MADRS	↓ MADRS (*p* < 0.001) Response rate: 29% Remission rate: 17%
Sagud et al. [49]	14	52.8 (±10.4)	Prospective, non-comparative, open-label	Quetiapine	Mean (SD): 315.0 (±109.0)	Various ADs	20 weeks	HAM-D total score and insomnia/depressive mood/anxiety subscales	↓ HAM-D total score (*p* < 0.001) ↓ HAM-D subscales (*p* < 0.001)
Papakostas et al. [48]	13	41.9 (±10.1)	Open-label	Ziprasidone	Mean (SD): 82.1 (±48.9)	SSRIs	6 weeks	HAM-D	Response rate: 61.5% Remission rate: 38.5%
Bauer et al. [54]	557	18–65	Head-to-head Open-label, R Comparison add-on QUE-XR/QUE-XR monotherapy/add-on LIT	Quetiapine	Mean (SD): add-on QUE-XR: 242.0 (±54.0) QUE-XR monotherapy: 238.0 (±60.0) add-on LIT: 882.0 (±212.0)	SSRIs or venlafaxine	6 weeks	MADRS	↓ MADRS: add-on QUE-XR = QUE-XR monotherapy = add-on LIT (*p* = 0.05)
Gobbi et al.	86	49.9 (±13.3)	Head-to-head Naturalistic Comparison SGAs (ARI, OLA, RIS, QUE)/another AD	Aripiprazole Olanzapine Risperidone Quetiapine	Mean (SD): ARI 4.4 (±1.3) OLA 8.7 (±1.8) RIS 1.88 (±0.5) QUE 129.0 (±29.0) ADs at various dosages	SSRIs or SNRIs	3 months	MADRS, HAM-D, QIDS, CGI	↓ MADRS, HAM-D, QIDS, CGI scores from baseline to endpoint in both SGA and ADs groups (*p* < 0.001)↓ depressive symptoms MADRS and HAM-D: SGA > ADs (*p* < 0.05)
Mohamed et al.	1137	54.4 (±12.2)	Head-to-head, R Comparison add-on ARI/add-on BUP/BUP switching	Aripiprazole	ARI max 10 BUP max 400	Various ADs	12 weeks	Remission at QIDS	Add-on ARI > BUP switching (*p* = 0.02) Add-on ARI = add-on BUP (*p* = 0.47) Add-on BUP = BUP switching (*p* = 0.09)

Key: * = reanalysis on the sample recruited by Fava et al. [35]. ** = subgroup analysis of “ADMIRE” by Kamijima et al. [39]. *** = subgroup analysis on the patients over 55 of the total sample recruited by Rapaport et al. [45]. ≈ = same as above. AD = antidepressant; ARI = aripiprazole; BREX = brexpiprazole; BREX-1 = brexpiprazole 1 mg/d; BREX-3 = brexpiprazole 3 mg/d; BUP = bupropion; CARI = cariprazine; CGI = Clinical Global Impression; CIT = citalopram; CLO = clomipramine; FLU = fluoxetine; HAM-D = Hamilton Depression Rating Scale; HD = high-dose; KSQ = Kellner Symptom Questionnaire; LD = low-dose; LIT = lithium; MADRS = Montgomery and Asberg Depression Rating Scale; MDD = major depressive disorder; OLA = olanzapine; PAR = paroxetine; pcb = placebo; QUE (XR) = quetiapine (extended release); QIDS = Quick Inventory of Depressive Symptomatology; R = randomized; RCT = randomized controlled trial; RIS = risperidone; SER = sertraline; SGA = Second generation antipsychotic; SNRI = Serotonin and Norepinephrine Reuptake Inhibitor; SSRI = Selective Serotonin Reuptake Inhibitor; TCA = tricyclic antidepressants.

**Table 3 ijms-22-13070-t003:** Summary of the studies about lithium augmentation in TRD.

Reference	*n*	Age Mean (SD), Years	Design	Dosage (mg/d)/Plasma Levels (mmol/L)	AD	Duration	Primary Outcome Measures	Results
**RCTs**								
Bauer et al. [60]	27	47.4 (±16.9)	I: open, acute treatment phase II: RC continuation phase	Mean (SD): 980.0 (±295.6)/0.65 (±0.14)	Various ADs	4 months	Relapse in phase II (HAM-D)	Relapse rate: Li < pcb (*p* = 0.02)
Bschor et al. [61] *	22	46.4 (±15.7)	Double-blind Follow-up maintenance phase study	N.A.	Various ADs	1 year	Recurrence (HAM-D)	Recurrence rate: Li = pcb (*p* = 0.49)
Joffe et al.	50	37.4 (±11.2)	Head-to-head Double-blind Comparison Li/T3/pcb	Mean: Li: 935.3/0.68 T3: 37.5 mcg	Desipramine, imipramine	2 weeks	HAM-D	↓ HAM-D: Li > pcb (*p* = 0.04) T3 > pcb (*p* = 0.02) Li = T3 (*p* > 0.05)
Kok et al.	32	71.9 (±7.8)	Augmentation vs. switch Double-blind comparison VFX/NOR Not remitted patients → Open-label comparison Li augmentation/switch to PHE/switch to TCA/switch to ECT	Mean (SD): Li: 586.0 (±86.0)/0.82 (±0.15) PHE: 53.0 (±8.0)	Venlafaxine, nortriptyline	12 weeks	MADRS	↓ MADRS: Li (*p* < 0.001) TCA (*p* < 0.01) PHE (*p* > 0.05) ECT (*p* > 0.05)
Nierenberg et al. [59]	35	37.2 (±8.3)	Double-blind	N.A./0.61 (range: 0.6–0.9)	Nortriptyline	6 weeks	HAM-D, CGI	↓ HAM-D and ↓ CGI: Li = pcb (*p* > 0.05)
Stein and Bernadt, [58]	34	47.2 (±19.5)	Double-blind Experimental group: Li 250 3 weeks → 750 6 weeks Controls: pcb 3 weeks → Li 250 3 weeks → Li 750 3 weeks	Dose/mean (SD) Experimental group: 250/0.25 (±0.12) 750/0.78 (±0.35) Controls: 250/0.25 (±0.15) 750/0.65 (±0.21)	TCAs	9 weeks	MADRS	↓ MADRS: Li 250 = pcb (*p* = 0.81) Li 750 > Li 250 (*p* = 0.009)
Yoshimura et al.	30	Li: 39.0 (±8.0) OLA: 42.0 (±7.0) ARI: 40.0 (±10.0)	Head-to-head Comparison Li/OLA/ARI	Mean (SD): Li: 458.0 (±103.0)/N.A. OLA: 7.0 (±5.0) ARI: 9.0 (±6.0)	Paroxetine	4 weeks	HAM-D	↓ HAM-D: Li = OLA = ARI (*p* < 0.001) Response—remission rates: Li 40–20% OLA 30–10% ARI 40–20%
Open studies								
Bertschy et al. [67]	13	45	Open-label	Mean (SD): N.A./0.75 (±0.12)	Venlafaxine	4 weeks	MADRS	↓ MADRS: Li = pcb (*p* = 0.20)
Buspavanich et al. [70]	Tot: 167 Geriatric: 22 Non-geriatric: 145	Tot: 48.3 (±13.9) Geriatric: 71.9 (±5.6) Non-geriatric: 44.8 (±11.0)	Prospective multicenter cohort study Comparison geriatric/non-geriatric patients	Mean (SD): 150.0 (±89.9)/0.68 (±0.2)	Various ADs	4 weeks	HAM-D	Response rate: geriatric > non-geriatric patients (*p* = 0.04)
Dallal et al. [63]	20	27–63 42.0 (±10.3)	Open-label	Range: 150–300/0.5–1.2	Desipramine	6 weeks	CGI	↓ CGI (*p* < 0.01)
Dinan [65]	11	37–59	Open-label	Dose/mean (SD): 400/0.26 (±0.1) or 800/0.6 (±0.1)	Sertraline	1 week	HAM-D	↓ HAM-D (*p* < 0.01) Response not related to Li plasma levels
Doree et al.	20	QUE: 52.3 (±8.1) Li: 49.3 (±9.4)	Head-to-head Open-label Comparison QUE/Li	Mean: QUE: 430 (range: 300–700) Li: N.A./0.78	Various ADs	8 weeks	HAM-D MADRS	↓ HAM-D both QUE and Li: *p* < 0.0001 QUE > Li (*p* < 0.05) ↓ MADRS: both QUE and Li: *p* < 0.0001 QUE > Li (*p* < 0.05)
Flint and Rifat [69]	21	64–88 75.6 (±7.1)	Prospective, open-label	Mean (SD): N.A./0.67 (±0.17)	Nortriptyline, fluoxetine, phenelzine	2 weeks	HAM-D, HAD	Response rate: 24%
Fontaine et al. [64]	60	24–55 FLU+Li: 42.6 (±8.6) DES + Li: 41.0 (±8.8)	Open-label Comparison FLU + Li/DES + Li	Mean (SD): FLU + Li: 570.0 (±120.8)/N.A. DES + Li: 660.0 (±165.3)/N.A.	Fluoxetine, Desipramine	6 weeks	CGI	↓ CGI: FLU + Li = DES + Li Response rate FLU + Li: 60% Response rate DES + Li: 56.6% Rapid response (1 week): FLU + Li > DES + Li (N.S.)
Gervasoni et al.	10	N.A.	Head-to-head Observational Comparison Li-s/Li-s + T3	Li-s: 1320/median: 0.52 (range: 0.38–1.10)T3: 37.5 mcg	Clomipramine	2 months	MADRS	Remission rates: Li-s = 10% Li-s + T3 = 0
Hoencamp et al. [66]	22	43.0 (±13.0)	Open-label	Dose/mean (SD): 600/0.66 (±0.19)	Venlafaxine	7 weeks	HAM-D MADRS CGI	↓ HAM-D (*p* = 0.001) Response rate: 34.8% Remission rate: 8.7% ↓ MADRS (*p* = 0.005) ↓ CGI (*p* = 0.001)
Ivkovic et al.	88	LAM: 54.2 (±13.7) Li: 49.3 (±12.3)	Head-to-head Open-label	Mean (SD): LAM: 117.7 (±54.3) Li: 900/N.A.	Various ADs	8 weeks	HAM-D, CGI	LAM = Li ↓ HAM-D (*p* = 0.83) ↓ CGI (*p* = 0.92) Within 2nd week: LAM > Li ↓ HAM-D (*p* = 0.01) ↓ CGI (*p* = 0.02)
Kok et al. [71]	28	Li: 73.6 (±7.3) PHE: 72.6 (±7.7)	Head-to-head Open-label, R Comparison Li augmentation/switch to PHE	Mean (SD) Li: 527.0 (±96.0)/0.71 (±0.17) PHE: 46.0 (±9.0)	TCA or venlafaxine	6 weeks	MADRS remission	↓ MADRS Li (*p* < 0.001) PHE (*p* = 0.85) Remission rates: Li > PHE (*p* = 0.04) Response rates: Li > PHE (*p* = 0.03)
Schindler and Anghelescu	34	Li: 50.3 (±13.6) LAM: 45.1 (±13.4)	Head-to-head Open-label, R Comparison Li/LAM	Mean: Li: N.A./0.71 LAM: 152.9	Various ADs	8 weeks	HAM-D	↓ HAM-D: Li = LAM (*p* = 0.11)
Schüle et al.	46	50.78 (±12.27)	Head-to-head Open-label Comparison MIR monotherapy/Li augmentation/CAR augmentation	Mean (SD): Li: 917.0 (±144.1)/0.71 (±0.13) CAR: 370.0 (±67.5)/32.4 (±8.16)	Mirtazapine	3 weeks	HAM-D	Response rates: MIR: 21.7% Li 53.8% CAR 20.0% Li > MIR (*p* = 0.05) CAR = MIR (*p* = 0.91)
Thase et al. [62]	20	40.4 (±9.9)	Open-label	Responders: N.A./0.56 (±0.22) Non-responders: N.A./0.83 (±0.19)	Imipramine	6 weeks	HAM-D	↓ HAM-D (*p* < 0.001) Response rate: 65%
Whyte et al.	Augmentation group: 53 Switch group: 12	Augmentation group: 76.2 (±5.7) Switch group: 78.8 (±7.2)	Augmentation vs. switch Open-label Comparison Li/NOR/BUP/switch to VFX	Max, median (range): Li: 300 (225–300)/0.5–0.7 NOR: 35 (10–50) BUP: 200 (50–400) VFX: 244 (150–300)	Paroxetine	Median (range), weeks: Li: 7 (1–22.3) NOR: 17 (0.2–28.7) BUP: 5.6 (0.3–30) VFX: 12 (8.7–12)	HAM-D	Response rates: Li: 43% NOR: 31% BUP: 45% VFX: 41.7%
Zimmer et al. [68]	13	74.1 (±8.2)	Open-label	Range/mean (SD): 300–450/0.65 (±0.20)	Nortriptyline	3 weeks	HAM-D	↓ HAM-D (*p* < 0.001)

Key: * = study including subjects without relapse during the 4-months continuation phase reported by Bauer et al. [60]. AD = antidepressant; ARI = aripiprazole; BUP = bupropion; CAR = carbamazepine; CGI = Clinical Global Impression; DES = desipramine; ECT = Electro-Convulsive Therapy; FLU = fluoxetine; HAD = Hospital Anxiety and Depression; HAM-D = Hamilton Depression Rating Scale; IMAO = Monoamine Oxidase Inhibitor; LAM = lamotrigine; Li = lithium; Li 250 = lithium at 250 mg/d; Li 750 = lithium at 750 mg/d; Li-s = lithium sulfate; MADRS = Montgomery and Asberg Depression Rating Scale; MIR = mirtazapine; N.A. = not available; N.S. = not statistically significant; NOR = nortriptyline; OLA = olanzapine; pcb = placebo; PHE = phenelzine; QUE = quetiapine; R = randomized; RC = randomized controlled; RCT = randomized controlled trial; SD = standard deviation; T3: triiodothyronine; TCA = tricyclic antidepressant; VFX = venlafaxine.

**Table 4 ijms-22-13070-t004:** Summary of the studies about augmentation with antiepileptic drugs in TRD.

Reference	*n*	Age Mean (SD), y	Design	Augmentation Molecule	Dosage	AD	Duration	Primary Outcome Measures	Results
**RCTs**									
Barbosa et al. [81]	15	30.2 (±8.4)	Double-blind	Lamotrigine	Max: 100 mg/d	Fluoxetine	6 weeks	HAM-D MADRS CGI	↓ HAM-D: LAM = pcb (*p* = 0.21) ↓ MADRS: LAM = pcb (*p* = 0.46) ↓ CGI: LAM > pcb (*p* = 0.03)
Santos et al. [82]	27	38.2 (±8.7)	Double-blind	Lamotrigine	Max: 200 mg/d	Various ADs	8 weeks	CGI MADRS	↓ CGI: LAM = pcb (*p* = 0.45) ↓ MADRS: LAM = pcb (*p* = 0.45; *p*-adj = 0.88) Response rates: LAM = pcb (*p* = 0.60)
Shapira et al. [86]	20	47.5 (±14.1)	Double-blind	Phenytoin	N.A.	Fluvoxamine, fluoxetine, paroxetine	4 weeks	HAM-D	↓ HAM-D: PHE = pcb (*p* = 0.30)
Mowla and Kardeh [89]	42	36.2	Double-blind	Topiramate	Range: 100–200 mg/d Mean: 173.15 mg/d	Fluoxetine, citalopram, sertraline	8 weeks	HAM-D CGI	↓ HAM-D TOP: *p* < 0.001 ↓ HAM-D and ↓ CGI: TOP > pcb (*p* < 0.001)
Fang et al. [91]	193	Range: 18–65	Head-to-head Multicenter, double-blind Comparison RIS/VAL/BUS/TRZ/T3	Valproate	RIS: 2 mg/d VAL: 600 mg/d BUS: 30 mg/d TRZ: 100 mg/d T3: 80 mg/d	Paroxetine	8 weeks	Remission at HAM-D	Remission rates: overall 37.3% RIS: 26.7% VAL: 48.7% BUS: 32.6% TRZ: 42.6% T3: 37.5% RIS = VAL = BUS = TRZ = T3 (*p* = 0.25)
Open studies									
Barbee and Jamhour [83]	31	50.2 (±11.2)	Retrospective	Lamotrigine	Mean: 112.9 mg/d	Various ADs	Mean 41.8 weeks (at least 6 weeks)	CGI	(Very) much improved: 48.4% Mildly improved: 22.6% Unchanged: 29.0%
Gutierrez et al. [85]	34	48.0 (±7.4)	Retrospective	Lamotrigine	Mean: 113.3 mg/d	Various ADs	1 year	Medication Visit by MD	↓ scores in target symptoms: - cognitive impairment, depressed mood, irritability, loss of interest (*p* < 0.01) energy (*p* < 0.001) sleep disturbance (*p* > 0.05)
Rocha and Hara [84]	25	Range: 18–65	Retrospective	Lamotrigine	Mean (SD): 155.0 (±64.5) mg/d	Various ADs	4 weeks	CGI	Response rate: 76%
Karaiskos et al. [87]	20	72.6 (±6.3)	Open-label	Pregabalin	Mean (SD): 106.0 (±78.0) mg/d	Various ADs	12 weeks	HAM-D HAM-A	↓ HAM-D: *p* < 0.01 ↓ HAM-A: *p* < 0.05
Ghabrash et al. [88] 2015	14	Range: 19–59	Comparison of psychometric scores at T0 (pre-treatment) with scores at: T1 (1 month) T4 (4 months) T7 (7 months)	Valproate	375–1000 mg/d	Various ADs	7 months	MADRS CGI	↓ MADRS: T0 vs. T1 (*p* < 0.001) T4 (*p* < 0.001) T7 (*p* < 0.001) ↓ CGI: T0 vs. T1 (*p* = 0.03) T4 (*p* < 0.001) T7 (*p* < 0.001)
Fornaro et al. [90]	24	50.7 (±0.2)	Open-label	Zonisamide	75 mg/d	Duloxetine	12 weeks	HAM-D	Response rate: 58.3%

Key: AD = antidepressant; BUS = buspirone; CGI = Clinical Global Impression; GAF = Global Assessment of Functioning; HAM-A = Hamilton Anxiety Rating Scale; HAM-D = Hamilton Depression Rating Scale; LAM = lamotrigine; MADRS = Montgomery–Åsberg Depression Rating Scale; MD = medical doctor; pcb = placebo; PHE = phenytoin; RCT = randomized controlled trial; RIS = risperidone; SD = standard deviation; SSRI = Selective Serotonin Reuptake Inhibitor; TOP = topiramate; TRZ = trazodone; VAL = valproate.

**Table 5 ijms-22-13070-t005:** Summary of the studies about augmentation with IV ketamine, esketamine nasal spray, and psychostimulant drugs in TRD.

Reference	*n*	Age Mean (SD), y	Design	Augmentation Molecule	Dosage	AD	Duration	Primary Outcome Measures	Results
**RCTs**									
Daly et al. [101] *	297	46.3 (±11.1)	Multicenter, double-blind TRD patients → 16 weeks ESK augmentation → R maintenance phase: comparison ESK/pcb	Esketamine	Remitters: 56 mg or 84 mg/2 weeks Responders: 56 mg or 84 mg once weekly	SSRIs or SNRIs	Median among remitters: 17.7 weeks Median among responders: 19.4 weeks	Time to relapse at MADRS	Relapse among remitters: pcb > ESK (*p* = 0.003) Relapse among responders: pcb > ESK (*p* < 0.001)
Fedgchin et al. [99]	315	46.3 (±11.2)	Multicenter, double-blind Comparison ESK56/ESK84/pcb	Esketamine	56 mg or 84 mg twice weekly	Escitalopram, Sertraline, Venlafaxine, Duloxetine	4 weeks	MADRS	↓ MADRS: ESK56 > pcb (*p* = 0.03) ESK84 = pcb (*p* = 0.09)
Ochs-Ross et al. [102]	122	70.0 (±4.52)	Double-blind	Esketamine	28 mg, 56 mg or 84 mg twice weekly	≈	≈	≈	↓ MADRS: ESK = pcb (*p* = 0.06) 65–74 years old: ESK > pcb (*p* = 0.02) ≥ 75 years old: ESK = pcb (*p* = 0.93)
Popova et al. [100]	197	ESK: 44.9 (±12.6) pcb: 46.4 (±11.1)	Multicenter, double-blind	Esketamine	56 mg or 84 mg twice weekly	≈	≈	≈	↓ MADRS: ESK > pcb (*p* = 0.02)
Fava et al., 2020 [94]	86	KETA 0.1: 43.1 (±11.9) KETA 0.2: 45.5 (±14.6) KETA 0.5: 48.6 (±12.9) KETA 1.0: 47.4 (±10.1)	Double-blind Comparison KETA 0.1-0.2-0.5-1.0 mg/kg/pcb	IV Ketamine	0.1-0.2-0.5-1.0 mg/kg	Various ADs	30 days	HAM-D6 at day 1 and 3	↓ HAM-D6: KETA > pcb (*p* = 0.03) KETA 0.1-0.2-0.5-1.0 mg/kg > pcb (*p* = 0.04) ↓ HAM-D6 day 1: KETA 0.1–0.2 mg/kg = pcb (*p*-adj = 0.14 and 0.79) KETA 0.5 mg/kg > pcb (*p*-adj < 0.001) KETA 1.0 mg/kg > pcb (*p*-adj = 0.04) ↓ HAM-D6 day 3: KETA 0.1-0.2-0.5-1.0 mg/kg = pcb (*p* > 0.05)
Freeman et al. [96] **	99	18–70 Males: 47.5 (±12.5) Females: 44.8 (±12.7)	≈	IV Ketamine	≈	≈	≈	≈	↓ HAM-D6: males = females (groupxgender *p* = 0.69)
Feeney et al. [98] **	56	45.7 (±12.3)	≈	IV Ketamine	0.1-0.5-1.0 mg/kg	≈	≈	MADRS suicide item	↓ MADRS suicide item at day 30: KETA > pcb (*p* = 0.03)
Ionescu et al. [95]	26	45.4 (±12.4)	Double-blind	IV Ketamine	0.5 mg/kg	Various ADs	3 weeks	HAM-D	↓ HAM-D: KETA = pcb (*p* = 0.47)
Salloum et al. [97] **	56	KETA 0.1: 47.0 (±8.1) KETA 0.5: 45.5 (±11.9) KETA 1.0: 45.3 (±9.6)	Double-blind Comparison KETA 0.1/0.5/1.0 mg/kg	IV Ketamine	0.1-0.5-1.0 mg/kg	Various ADs	30 days	MADRS	At day 3: Response rate: 48% Remission rate: 34% At day 30: Remission rate: 21%
Price et al. [104]	15	50.0 (±12.0)	Pcb substitution Comparison FEN/pcb	Fenfluramine (Amphetamine)	89.0 (±26.0) mg/d	Desipramine	Mean (SD): 16.4 (±5.0) d	SCRS HAM-D	↓ SCRS and ↓ HAM-D: FEN = pcb (*p* > 0.05)
Richards et al. [105]	Study 1: pcb = 201, LDX = 201; Study 2: pcb = 213, LDX = 211	Study 1 LDX: 42.2 (±12.3) Study 2 LDX: 42.0 (±11.6)	Multicenter, double-blind Comparison LDX/pcb	Lisdexamfetamine dimesylate (Amphetamine)	Study 1: 46.5 (±13.7) mg/d Study 2: 43.4 (±14.3) mg/d	Various ADs	16 weeks	MADRS	↓ MADRS Study 1: LDX = pcb (*p* = 0.88) ↓ MADRS Study 2: LDX = pcb (*p* = 0.58)
Patkar et al. [106]	50	48.5	Double-blind	Metilphenidate	34.2 (±6.3) mg/d	Various ADs	4 weeks	HAM-D	↓ HAM-D: MPH = pcb (*p* = 0.22) Response rates: MPH > pcb (*p* = 0.12)
Ravindran et al. [107]	134	43.8 (±11.0)	Multicenter, double-blind	Metilphenidate	36.4 (±9.1) mg/d	Various ADs	5 weeks	MADRS	↓ MADRS: MPH = pcb (*p* = 0.74)
Open studies									
Wajs et al. [103] ***	150	52.2 (±13.7)	Multicenter, open-label	Esketamine	Flexible 14–84 mg once weekly or every-other-week	≈	1 year	≈	Response rate: 76.5% Remission rate: 58.2%
Cusin et al. [93]	12	48.9	Open-label 6 infusions (2/week): infusion 1–3: 0.50 mg/kg (IV); infusion 4–6: 0.75 mg/kg (IV)	IV Ketamine	Mean (SD): infusion 1–3: 29.0 (±16.2) mg infusion 4–6: 43.5 (±24.3) mg	Various ADs	3 weeks	HAM-D	↓ HAM-D: *p* < 0.001 Response rate: 41.7% Remission rate: 16.7%
Schiroma et al. [92]	14	54.0	Open-label	IV Ketamine	0.5 mg/kg	Various ADs	12 days	MADRS	↓ MADRS (*p* < 0.001)
Nasr et al. [108]	78	44.0	Retrospective	Modafinil	249.0 (±122.0) mg/d	Various ADs	9 months	CDRS	↓ CDRS: *p* < 0.01

Key: * = analysis in the long-term of patients who achieved response in the studies by Fedgchin et al. [99] and Popova et al. [100]. ** = reanalysis on a subgroup of patients recruited by Fava et al. [94]. *** = analysis in the long-term, including, in the total sample, patients who completed the study by Ochs-Ross et al. [102]. ≈ = same as above; AD = antidepressant; CDRS = Carroll Depression Rating Scale; ESK = esketamine nasal spray; ESK56 = esketamine at 56 mg/d; ESK84 = esketamine at 84 mg/d; FEN = fenfluramine; HAM-D = Hamilton Depression Rating Scale; HAM-D6 = Hamilton Depression Rating Scale—6 items; IV = intravenous; KETA = ketamine; LDX = lisdexamfetamine dimesylate; MADRS = Montgomery and Asberg Depression Rating Scale; MPH = metilphenidate; *p*-adj = *p* value adjusted for multiple comparisons; pcb = placebo; RCT = randomized, controlled trial; SCRS = Short Clinical Rating Scale; SD = standard deviation.

**Table 6 ijms-22-13070-t006:** Summary of the studies about augmentation with non-psychopharmacological agents, other molecules and supplements in TRD.

Reference	*n*	Age (Years)	Design	Augmentation Molecule	Dosage	AD	Duration	Primary Outcome Measures	Results
**RCTs**									
McAllister-Williams et al. [110]	165	Range: 18–65	Double-blind	Metyrapone	500 mg/bid	Various ADs	3 weeks	MADRS	↓ MADRS: MET = pcb (*p* = 0.74)
Nettis et al. [112]	39	MIN (*n* = 18): 47.0 (±10.0) pcb (*n* = 21): 43.7 (±10.7)	Double-blind Comparisons: MIN vs. pcb CRP+/MIN vs. CRP-/MIN vs. CRP+/pcb vs. CRP-/pcb	Minocycline	200 mg/d	≈	4 weeks	HAM-D	↓ HAM-D: - MIN = pcb (*p* = 0.13) - CRP+/MIN > other subgroups (*p* < 0.001)
Mischoulon et al.	12	Range: 18–65	Double-blind	Naltrexone	1 mg/bid	Dopaminergic agents	3 weeks	≈	↓ HAM-D: LNT = pcb (*p* = 0.30)
Moreno et al. [113]	10	Mean (SD): 43.0 (±13.0)	Double-blind, crossover	Pindolol	2.5 mg/tid	Desipramine, fluoxetine, bupropion	2 weeks	≈	↓ HAM-D: PIN = pcb (*p* = 0.72)
Perez et al.	78	Mean (SD): 47.1 (±10.1)	Double-blind	Pindolol	2.5 mg/tid	Clomipramine, fluoxetine, fluvoxamine, paroxetine	10 days	≈	↓ HAM-D: PIN = pcb (*p* = 0.22) Remission rates: PIN = pcb (*p* > 0.05)
Sokolski et al. [114]	9	N.A.	Double-blind	Pindolol	7.5 mg	Paroxetine	4 weeks	≈	*↓* HAM-D: PIN > pcb *(p =* 0.001)
Perry et al.	34	Mean (SD): PIN: 49.0 (±13.0) pcb: 43.0 (±11.0)	Double-blind, hemi-crossover	Pindolol	2.5 mg/bid	Fluoxetine, paroxetine, sertraline	6 weeks	≈	*↓* HAM-D: PIN = pcb *(p* = 0.93) ↓ HAM-D core mood item: PIN = pcb (*p* = 0.50)
Price et al.	8	Mean (SD): 50.5 (±13.2)	Double-blind	Reserpine	5 mg/bid (IM)	Desipramine	12 days	SCRS, HAM-D	↓ SCRS and ↓ HAM-D: RES = pcb (*p* > 0.05)
Targum et al.	234	Range: 21–69 Mean (SD): 47.2 (±10.78)	Multicenter, double-blind	SAME	800 mg/d	Various ADs	6 weeks	HAM-D, MADRS, IDS-SR30	SAME = pcb: ↓HAM-D (*p* = 0.83) ↓MADRS (*p* = 0.42) ↓IDS-SR30 (*p* = 0.70)
Dichtel et al.	87 women	Range: 21–70 Mean (SD): 47.0 (±14.0)	Double-blind	Testosterone	Mean (SD): 12.2 mg/d (±5.6)	SSRIs or SNRIs	8 weeks	MADRS	↓ MADRS: TXT = pcb (*p* = 0.91)
Siwek et al.	21	Range: 18–55 Mean (SD): 46.2 (±5.8)	Double-blind	Zinc	25 mg/d	Imipramine	12 weeks	HAM-D, MADRS, BDI, CGI	Zinc > pcb: ↓ HAM-D (*p* < 0.02) ↓ MADRS (*p* < 0.01) ↓ BDI (*p* < 0.02) ↓ CGI (*p* < 0.01)
Open studies									
Mendlewicz et al. [109]	17	Range: 29–62 Mean (SD): 46.1 (±9.7)	Open-label	ASA	160 mg/d	SSRIs	4 weeks	HAM-D	↓ HAM-D (*p* < 0.0001) Response rate: 52.4% Remission rate: 43%
Avari et al. [111]	13	73.1 (±11.2)	≈	Minocycline	100 mg twice/d	Various ADs	8 weeks	MADRS	↓ MADRS Remission rate: 31%
Miller et al.	9 women	Range: 25–59 Mean (SD): 48.1 (±12.2)	≈	Testosterone	300 mcg/d (transdermal)	SSRIs or SNRIs	≈	≈	↓ MADRS (*p* = 0.004)
Rudas et al.	9	Range: 21–68	≈	T3/T4	Range: 150–300 mcg/d Mean (SD): 235.0 (±58.0) mcg/d	Various ADs	≈	HAM-D	↓ HAM-D (*p* < 0.01)
Hori and Kunugi	12	18–64	≈	Pramipexole	0.25–3 mg/d	≈	12 weeks	HAM-D, CGI	↓ HAM-D (*p* < 0.0001) ↓ CGI (*p* = 0.003)
Cassano et al.	7	18–74	Prospective, open-label	Ropinirole	0.25–1.5 mg/d	≈	16 weeks	MADRS, CGI	↓MADRS (*p* < 0.02) Response rate: 40% Remission rate: 40%
De Berardis et al.	25	Mean (SD): 32.0 (±5.1)	Open- label, single-blind	SAME	800 mg/d	≈	8 weeks	HAM-D	↓ HAM-D (*p* < 0.001) Response rate: 62.5% Remission rate: 37.5%

Key: ≈ = same as above; AD = antidepressant; ASA = acetylsalicylic acid; BDI = Beck Depression Inventory; bid = bis in die; CGI = Clinical Global Impression; d = day; HAM-D = Hamilton Depression Rating Scale; IDS-SR30 = Self-rated Inventory of Depressive Symptomatology; IM = intramuscular; LNT = low-dose naltrexone; MADRS = Montgomery and Asberg Depression Rating Scale; MET = metyrapone; MIN = minocycline; pcb = placebo; PIN = pindolol; RCT = randomized controlled trial; RES = reserpine; SAME = S-AdenosylMethionine; SCRS = Short Clinical Rating Scale for Depression; SD = standard deviation; SNRI = Serotonin-Norepinephrine Reuptake Inhibitor; SSRI = Selective Serotonin Reuptake Inhibitor; tid = ter in die; TXT = testosterone.

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
