# Peer review of "Augmentative Pharmacological Strategies in Treatment-Resistant Major Depression: A Comprehensive Review"

_ijms, 2021, doi:10.3390/ijms222313070_

Round 1
Reviewer 1 Report
This review addresses an important topic of high clinical relevance. Methods are timely and appropriate. Results are clearly presented in text and tables. Discussion is appropriate. The manuscript is well written.
I have only a few minor comments.
line (l) 108 and Figure 1- "different language "/"No English language" is reported as reason of exclusion. This citerion is not mentionned in the Materiala and Methods section. Please add it there.
- 112 - please change "defined" to "define".
- l 134 - please change "showed" to "show".
- l 179 - please change "showed" to "shown".
- l 492 - "authors" in lower case please.
Author Response
First of all, we would like to thank the reviewer for the interest in the manuscript and the useful suggestions aimed at improving the manuscript. We added the criterion in the Methods section and we reported the suggested changes.
Reviewer 2 Report
The review article entitled " Augmentative pharmacological strategies in Treatment-Resistant Major Depression: a systematic review ". The Caldiroli et al., worked on the systematic review and identified the available literature about pharmacological augmentation to Antidepressants (AD) in Treatment Resistant Depression (TRD). According to the authors report, aripiprazole and lithium may be considered by clinicians as potential effective augmentative strategies in TRD. In addition, present systematic review was to update the available data about the efficacy of different pharmacological strategies as augmentation in TRD, in order to guide the clinician through the complex choice of the most adequate pharmacological treatments.
The manuscript comprises all the necessary elements of scientific novelty. The manuscript is well written and substantiated with detailed reports. I recommend this article for publication after incorporating minor changes given in below.
Line 25: Need to be reframed.
Authors must concentrate on the formatting, and use of symbols, etc., There is no uniformity was observed in the manuscript.
Discussion and conclusion section looks shallow. It needs to be improved. Discuss more and it will be useful to the readers for ease of understanding.
Abstract need to be modified.
Modify the figure 1 as more interactive and color.
Author Response
First of all, we would like to thank the reviewer for the interest in the manuscript and the useful suggestions aimed at improving the manuscript.
Unfortunately, we can't identify the line 25 in our version of the manuscript. We kindly ask the reviewer to speficify which sentence needs to be rephrased.
We checked the manuscript for uniformity of the symbols and format.
We improved the discussion as the reviewer suggested, adding an "expert opinion" paragraph to make the conclusions easier to be understood.
We kindly ask the reviewer to specify which kind of modifications the abstract needs.
We modified Figure 1, adding colours, as the reviewer suggested.
Reviewer 3 Report
Submitted review concerns an important aspect of modern psychiatry i.e. the possibility of enhancement of antidepressant drugs' action by the concomitant treatment with other agents. Despite my appreciation for the concept of presenting such data in an organized way, unfortunately, I believe that authors have missed a number of important research papers due to their restricted inclusion criteria.
Major comments:
1) The search was carried out only for "augmentation AND major depression" yielding approx. 1500 citations, whereas e.g. a search including : "augmentation OR adjunct* AND major depression" yields approx. 4100 citations.
Therefore, I am afraid, the presented version of review cannot be considered systematic. However, it can be submitted as a regular review paper, with an appropriate change of text.
2) Furthermore, at the present form, the discussion is mostly a summary of results. It does not really indicate what are the most promising ways, in the opinion of authors, to enhance the antidepressant action.
Author Response
First of all, we would like to thank the reviewer for the interest in the manuscript and the useful suggestions aimed at improving the manuscript.
1) thank you for the observation. We changed the text according to your suggestion, avoiding the "systematic" definition of the review paper
2) We modified the discussion adding an "expert opinion" paragraph, in order to clarify the highlights of the manuscript and the opinion of authors
Round 2
Reviewer 3 Report
At present, the manuscript is suitable for publication.